# SMU-Net: Style matching U-Net for brain tumor segmentation with missing modalities

**Reza Azad**                               Reza.Azad@lfb.rwth-aachen.de
**Nika Khosravi**                        Nika.Khosravi@rwth-aachen.de
**Dorit Merhof**                         Dorit.Merhof@lfb.rwth-aachen.de

*Institute of Imaging and Computer Vision, RWTH Aachen University, Germany*

## Abstract

Gliomas are one of the most prevalent types of primary brain tumors, accounting for more than 30% of all cases and they develop from the glial stem or progenitor cells. In theory, the majority of brain tumors could well be identified exclusively by the use of Magnetic Resonance Imaging (MRI). Each MRI modality delivers distinct information on the soft tissue of the human brain and integrating all of them would provide comprehensive data for the accurate segmentation of the glioma, which is crucial for the patient's prognosis, diagnosis, and determining the best follow-up treatment. Unfortunately, MRI is prone to artifacts for a variety of reasons, which might result in missing one or more MRI modalities. Various strategies have been proposed over the years to synthesize the missing modality or compensate for the influence it has on automated segmentation models. However, these methods usually fail to model the underlying missing information. In this paper, we propose a style matching U-Net (SMU-Net) for brain tumour segmentation on MRI images. Our co-training approach utilizes a content and style-matching mechanism to distill the informative features from the full-modality network into a missing modality network. To do so, we encode both full-modality and missing-modality data into a latent space, then we decompose the representation space into a style and content representation. Our style matching module adaptively recalibrates the representation space by learning a matching function to transfer the informative and textural features from full-modality path into a missing-modality path. Moreover, by modelling the mutual information, our content module surpasses the less informative features and re-calibrates the representation space based on discriminative semantic features. The evaluation process on the BraTS 2018 dataset shows the significance of the proposed method on the missing modality scenario.

**Keywords:** Missing modalities, brain tumor, content-style matching, segmentation.

## 1. Introduction

Magnetic resonance imaging (MRI) is a technique for visualizing the human brain's 3D structure and anatomical localization. This assistive method provides comprehensive information on human brain tissues for prognosis, diagnosis and as a result, selecting the most effective followup medical treatment. Analyzing the captured MRI scans requires an expert-level annotation (e.g. determining brain tumor region), which is usually costly and time-consuming. Moreover, radiologist annotation almost always comes with uncertainty due to limitations and artifacts in an imaging modality. To tackle this limitation, a set of complementary modalities is usually considered, such as T1, T1c, T2, and Flair. The main

objective of multi-modality data is to provide interrelated information for the radiologist to reduce the inherent uncertainty in the imaging reconstruction. It is critical to carefully analyze each of these modalities as this provides structural and quantitative information for brain tumor subregions annotation (Azad et al., 2022; Reza et al., 2022).

Early approaches to solve the missing modalities problem proposed synthesizing the missing modality. According to (Van Tulder and de Bruijne, 2015), using a synthesis method, which is more adaptable than the classifier, to infer missing data at test time may enhance multi-modal image classification by providing transformations of the data for the classifier and also expanding the training set. Random forest as a simple classifier as discussed in (Van Tulder and de Bruijne, 2015), has been demonstrated to enhance segmentation using synthesized data. (Jog et al., 2017) offers a random forest non-linear regression approach for synthesizing the missing modalities, which is capable of synthesizing T2 and FLAIR, which was previously thought to be a hurdle. However, (Van Tulder and de Bruijne, 2015) demonstrated through experiments that substituting the missing modality with a sequence created by a three-layered feed forward neural network yielded the same outcomes as replacing it with zeros or performing the segmentation without it. As a consequence, synthesizing or imputing the data will not worsen the results; they may, however, result in no improvement in some circumstances. Furthermore, such methods are applied to the input level and usually are limited in representative power to reconstruct the missing modality. To overcome this constraint, it is favorable to use deep learning as a powerful tool.

U-HeMIS (Havaei et al., 2016) is among the first approaches which proposed to incorporate the missing modality inside the segmentation model. In their design, a mapping function (regular encoder model) from each modality into an embedding space is learned. Then the first and second order moments (mean and variance) are used to model the common representation space shared between the modalities. Although U-HeMIS strives to create a shared latent embedding for all modalities, calculating the mean and variance will not necessarily reconstruct the missing information. Further efforts to build a common latent space representation resulted in the development of the HVED (Dorent et al., 2019). This approach utilizes independent encoders to extract first order moment features from each modality. Then it forms a common representation space by modeling the Gaussian distribution over the encoded features. Even though this shared latent space provides a common representation for all modalities, it fails to model the modality-dependent features and usually performs poorly when more than one modality is missing.

Other approaches, such as the network described in (Zhou et al., 2021a), take advantage of the significant correlation, which each two modalities share, and perform the segmentation task without a perfect reconstruction of the missing sequence. Similarly, (Zhou et al., 2021b) is a latent correlation representation learning method for addressing the missing modality problem that is not regarded a fully fulfilling method for all modalities. Later approaches such as (Wang et al., 2021) and (Vadacchino et al., 2021) proposed hierarchical adversarial knowledge distillation networks. However, these methods usually fail to reconstruct the textural (style) information from the missing modalities.

To overcome the aforementioned limitation, we propose SMU-Net for brain tumor segmentation with missing modalities. Our network takes into account the strength of the co-training strategy to distill the useful information from the full-modality model into a missing modality network, and thus, is capable of recovering the missing information. In

our design, we decompose the representation space into content and style representations. Then, by utilizing the matching modules we reconstruct the missing information. Our style matching module learns a non-linear parametric function to map the texture of full-modality into a missing modality network and ensures the reconstruction of informative features. Furthermore, the content module uses a context loss to surpass the less informative features and encourages discriminative feature learning. Our contributions are as follows:

- Model the common representation space using style-content matching modules

- Co-training approach to distill the informative features from the full-modality into a missing modality network

- End-to-end training strategy with state-of-the-art results results

- Publicly available implementation source code (will be available after publication)

## 2. Proposed Method

Inspired by recent work on the co-training approach (Qiao et al., 2018) and style transfer networks (Ecker et al., 2015), we introduce SMU-Net (Figure 1), a style matching U-Net for brain tumor segmentation with missing modalities. Our network takes into account the strength of the co-training strategy to distill the useful information from the full-modality model into a missing modality network. In our design, we decompose the representation space into content and style representations, then, by utilizing the matching modules, we reconstruct the missing information. Our style-matching module learns a non-linear parametric function to match the shallow to deep features between the full and missing modality networks. Furthermore, by modelling the mutual information, our content module surpasses the less informative information and re-calibrates the representation space based on common characteristics. We will elaborate on each module in more detail.

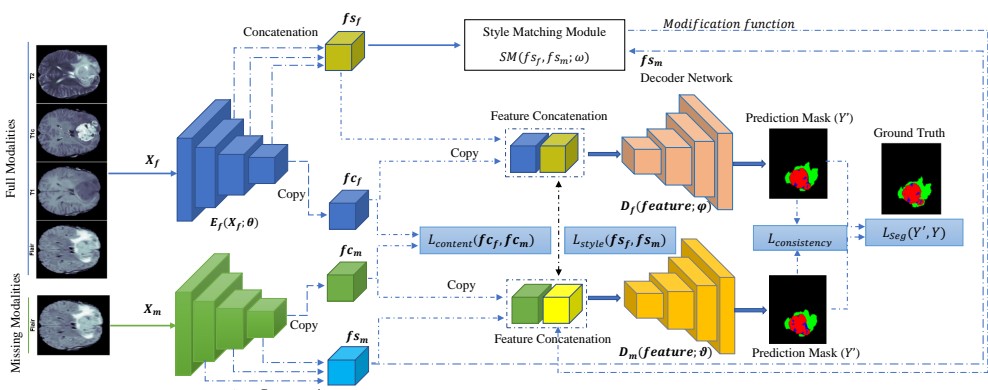

Figure 1: SMU-Net with 1) style matching mechanism to reconstruct the missing information using full-modality network and 2) content matching module to capture informative features and surpass the less discriminative ones.

## 2.1. Co-training Network

Previous efforts, such as the technique provided in (Qiao et al., 2018), have introduced the use of a co-training procedure for training deep learning models. Given that each MRI modality delivers useful information and reveals distinct characteristics about the soft tissue of the human brain, combining all the obtained characteristics provides us with comprehensive information for a thorough investigation of the brain's region of interest. In order to establish a training process that is resilient to missing modalities, our proposed network takes advantage of a coupled learning process. In our design there are two separate learning paths, 1) a path with the full modalities as input, and 2) a path in which the inputs come with missing modalities. The main objective of this design is to distill the informative features from the full-modality path into a missing-modality path, where the co-training strategy encourages the missing modality network to reconstruct the missing information. Following the standard notation in the segmentation task, we define the full-modality data as $X_f = \{\forall X_i, i \in M\}$, the missing modality version as $X_m = \{\forall X_i, i \in M'\}$, and the corresponding ground truth mask as $Y \in \{0, 1, 2, 4\}^{\{H \times W \times D\}}$. In our setting, the $X_i^{\{H \times W \times D\}}$ denotes a 3D modality image, with spatial dimension $(H, W, D)$ and $M$ shows the set of modalities (T1, T1w, T2 and Flair) where $M' \subset M$. The purpose of our co-training strategy is to distill the semantic and informative features from the full-modality model into a missing modality one. Thus, we define two parallel segmentation models (U-Net) which are trained independently and simultaneously using $X_f$ and $X_m$, respectively. Each network uses a Dice loss between the predicted mask $Y'$ and the ground truth $Y$ to learn the segmentation map, $\mathcal{L}_{seg} = \mathcal{L}_{dice}(Y'_f, Y) + \mathcal{L}_{dice}(Y'_m, Y)$. Next, we define the cross model consistency loss on both local and global levels to ensure feature distillation from the full-modality into a missing modality one. To this end, we calculate the mutual information between output (soft logits $(SL)$: network output before applying the softmax operation) of full and missing modality paths by using the Jensen-Shanon estimator (Sylvain et al., 2020):

$$\mathcal{L}_I(SL_f, SL_m) = \mathbb{E}_{\mathbb{P}_{SL_f SL_m}}\left[-\operatorname{sp}\left(CT_\phi(sl_f, sl_m)\right)\right] - \mathbb{E}_{\mathbb{P}_{SL_f} \otimes \mathbb{P}_{SL_m}}\left[\operatorname{sp}\left(CT_\phi(sl_f, sl_m)\right)\right], \quad (1)$$

where $\operatorname{sp}(z) = \log\left(1 + e^z\right)$ and $CT_\phi$ shows our co-training network with parameters $\phi$. This mutual maximization function ensures the distribution matching between both paths. To further include the global representation matching we calculate the $L1$ distance between soft logits of the full and missing modality paths:

$$\mathcal{L}_{L1}(SL_f, SL_m) = \sum_{i=1}^{c} |GP(SL_f) - GP(SL_m)|, \quad (2)$$

where $GP$ is the global pooling operation and $c$ is the number of classes. Our consistency loss comprises these these two metrics: $\mathcal{L}_{consistency} = \mathcal{L}_I + \mathcal{L}_{L1}$. The suggested cross-modal consistency loss evaluates the distribution matching between the outputs produced by the missing and full modality paths to improve output consistency by taking into account their correlations and eventually boosting the two paths to yield the same distribution. Besides that matching, both local and global level distribution ensures the minimum distribution distance between two paths and encourages the full modality path to supervise the missing modalities path by distilling the informative features. To further ensure the feature

distillation in both the shallow parts and bottleneck of the network, we propose the style and content matching modules, where the style matching function aims to reconstruct the missing information using the full-modality distribution while the content matching module tries to learn a discriminative representation and surpass the less informative ones.

## 2.2. Style-matching Module

Each MRI-modality is acquired using a different combination of properties and settings, and each one exposes distinct anatomical structures of the human brain. In terms of style information such as texture, contrast, saturation, and so on, the obtained MRI-modalities may differ, and this style variance exacerbates the domain shift problem across the full and missing modality paths. When the domain shift issue arises, the performance of most deep learning models in the missing modality scenario degrades. Therefore, we propose a style-matching module that can overcome the domain shift and, consequently, recover lost style information across full-modality and missing-modality paths. To this end, we decompose the feature representation into a style and a content representation of the input images. For the style representation, we concatenate the convolutional filter responses in shallow and deep layers whereas for the content one we use the last convolutional output. The derived style representation maintains valuable textural information, whereas the content feature contains the image's core structural and semantic characteristics. To perform the matching mechanism, we define three different style matching modules as depicted in Figure 2.

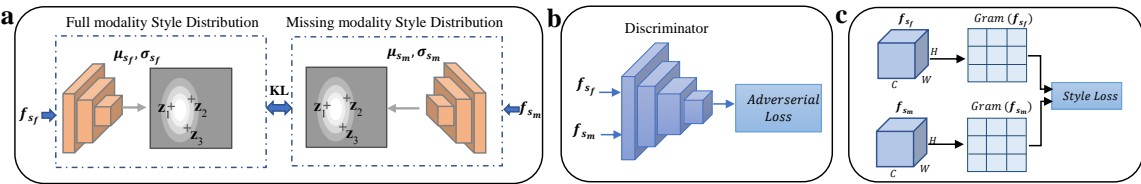

Figure 2: Style matching module comprising a) distribution matching, b) adverserial learning, and c) texture matching.

### 2.2.1. DISTRIBUTION MATCHING

In our first design, we distill the extracted style features from both paths into a continuous distribution space and then strive to minimize the Kullback-Leibler (KL) divergence to quantify the matching between two style distributions (Figure 2a):

$$\mathcal{L}_{\text{style}}^{kl} \ (fs_f, fs_m) = D_{\text{KL}}(T(z \mid fs_f, fs_m) || P(z \mid fs_f)), \tag{3}$$

where $z = (\theta, \sigma)$ is the parameters of the continuous distribution. By minimizing the KL loss, we align the distribution of the missing modality to follow the full-modality version, and thus, it learns to recover the missing information by learning the posterior distribution.

### 2.2.2. Adversarial learning

As illustrated in Figure 2 b), our second suggested style-matching mechanism is an adversarial learning module. Since exact segmentation is highly reliant on high-level representations, we want to try to prevent feature distribution misalignment in latent space. To this end, we use a discriminator network D parametrized with $\vartheta$ to discriminate full and missing modality styles. By performing an adversarial loss we genuinely align the feature distributions of the missing modality to the full-modality to recapture the missing information.

$$\mathcal{L}_{style}^{adv}\left(fs_f, fs_m\right) = \log\left(1 - D_\vartheta\left(fs_f\right)\right) + \log\left(D_\vartheta\left(fs_m\right)\right) \tag{4}$$

### 2.2.3. Texture matching

In the third approach we model the style matching module by maximizing the correlation between the texture of both representations. To this end, we compute the mean-squared error across the elements of the Gram matrices, which are created for each path. The weighted MSE $(w_l)$ loss of all layers is computed as follows (Ecker et al., 2015):

$$\mathcal{L}_{\text{style}}^{texture}\left(fs_f, fs_m\right) = \sum_{l=0}^{L} w_l \frac{1}{4C_l^2 N_l^2} \sum_j \left(fs_f^{l,j} - fs_m^{l,j}\right)^2, \tag{5}$$

where $N$ shows the spatial dimension and $C_l$ is the number of channels in the layer $l$. We should note that this loss function is consistently used in conjunction with the other style modules to approximate the style of missing modality with the full one.

## 2.3. Content Module

The content module in our design aims to capture the structural and contextual information shared among modalities. To distill this discriminative information from the full-modality path into a missing version we deploy the MSE loss between content representations:

$$\mathcal{L}_{\text{content}}\left(\boldsymbol{fc_f}, \boldsymbol{fc_m}\right) = \frac{1}{2} \sum_j \left(fc_{f_j} - fc_{m_j}\right)^2. \tag{6}$$

The content matching module guides the missing version to capture the discriminate features while surpassing the less informative ones. Once the style and content features are recalibrated we recombine these features, along with the ones extracted by the encoder, to generate a new representation as an input for the decoder module. We simply use a U-Net decoder to generate the segmentation map.

## 2.4. Joint Objective

The overall loss function optimized during the training consists of four terms:

$$\mathcal{L}_{\text{joint}} = \lambda_1 \mathcal{L}_{\text{segmentation}} + \lambda_2 \mathcal{L}_{\text{consistency}} + \lambda_3 \mathcal{L}_{\text{style}} + \lambda_4 \mathcal{L}_{\text{content}}. \tag{7}$$

We use $\lambda$ coefficients to weight the contribution of each loss on the overall loss value. We train the model in an end-to-end manner using Adam optimization with a learning rate $10^{-4}$ and batch size 1 for 250 epochs. It is worthwhile to mention that the code is developed in the Pytorch library and is carried out on a single RTX 3090 GPU.

## 3. Experimental Results

### 3.1. Dataset and Evaluation Metrics

For the evaluation process we use the publicly available BraTS 2018 dataset from the Multimodal Brain Tumor Segmentation Challenge (BraTS 2018) challenge (Menze et al., 2014; Bakas et al., 2017, 2018) . This dataset includes MRIs of a total of 285 patients: 210 gliomas with high grade and 75 gliomas with low grade, each of which containing the four aforementioned modalities (T1, T1c, T2, FLAIR). Each image's ground truth segmentation in the BraTS 2018 dataset includes labeling for four tissue classes: necrosis, edema, non-enhancing tumor, and enhancing tumor. Despite the fact that four distinct tumor labels are provided, they could well be categorized into three subregions for evaluation: the whole tumor (WT), the core tumor (CT), and the enhancing tumor (ET). Our pre-processing and data division follow the (Dorent et al., 2019) setting with input size $160 \times 192 \times 128$. For the evaluation, we use the Dice similarity coefficient (DSC). We have provided the implementation code in github.

### 3.2. Comparison

Table 1 presents the performance of our proposed SMU-Net on the BraTS 2018 dataset. First, to demonstrate the effectiveness of the SMU-Net, we start by comparing it to the well-known U-HeMIS (Havaei et al., 2016) and HVED (Dorent et al., 2019) and recent ACN (Wang et al., 2021) architectures. We can observe that when more than one modality is missing, the performance of the U-HeMIS or HVED largely degrades, whereas SMU-Net achieves better results especially in single modality cases, as shown in Table 1. This observation reveals the importance of our style-content matching modules for capturing the missing information and re-calibrating the missing modality path. Moreover, we can also observe that the proposed method outperforms the SOTA methods by large margins (20%) in single modality scenarios, which shows the effectiveness of our co-training strategy. Besides, our method comparatively achieved better results than the competitor ACN method in our similar setting. To highlight the efficacy of our approach, we offer qualitative results of our model on a single modality case in Figure 3. It can be observed that for every single modality the method produces acceptable segmentation results, where these results were not feasible from input images without reconstructing the missing information. This clearly shows the importance of our co-training strategy in recovering the discriminative information.

### 3.3. Ablation Study

The quantitative findings for each of our suggested modules in the SMU-Net architecture are included in Table 2. First, we can observe that the proposed modules considerably improve the results for the missing-modality path's performance, and they allow the network to transfer essential knowledge from the full-modality path to the missing-modality path. Second, our design choice for the style-matching module shows that using the adversarial method provides a better reconstruction function compared to the distribution and texture approaches. It is also worthwhile to mention that applying the style matching module without content separation results in a performance reduction of about $1 - 2\%$.

Table 1: Performance comparison of the proposed SMU-Net on the BraTS 2018 dataset using Dice metric. Note our method uses adversarial style matching module.

| Modalities | | | | Complete | | | | Core | | | | Enhancing | | | |
|---|---|---|---|---|---|---|---|---|---|---|---|---|---|---|---|
| Flair | T1 | T1c | T2 | U-HeMIS | HVED | ACN | SMU-Net | U-HeMIS | HVED | ACN | SMU-Net | U-HeMIS | HVED | ACN | SMU-Net |
| ○ | ○ | ○ | ● | 79.2 | 80.9 | 85.4 | **85.7** | 50.5 | 54.1 | 66.8 | **67.2** | 23.3 | 30.8 | 41.7 | **43.1** |
| ○ | ○ | ● | ○ | 58.5 | 62.4 | 79.8 | **80.3** | 58.5 | 66.7 | 83.3 | **84.1** | 60.8 | 65.5 | 78.0 | **78.3** |
| ○ | ● | ○ | ○ | 54.3 | 52.4 | **78.7** | 78.6 | 37.9 | 37.2 | **70.9** | 69.5 | 12.4 | 13.7 | 41.8 | **42.8** |
| ● | ○ | ○ | ○ | 79.9 | 82.1 | 87.3 | **87.5** | 49.8 | 50.4 | 66.4 | **71.8** | 24.9 | 24.8 | 42.2 | **46.1** |
| ○ | ○ | ● | ● | 81.0 | 82.7 | 84.9 | **86.1** | 69.1 | 73.7 | 83.2 | **85.0** | 68.6 | 70.2 | 74.9 | **75.7** |
| ○ | ● | ● | ○ | 63.8 | 66.8 | 79.6 | **80.3** | 64.0 | 69.7 | 83.9 | **84.4** | 65.3 | 67.0 | **75.3** | 75.1 |
| ● | ● | ○ | ○ | 83.9 | 84.3 | 86.0 | **87.3** | 56.7 | 55.3 | 70.4 | **71.2** | 29.0 | 24.2 | 42.5 | **44.0** |
| ○ | ● | ○ | ● | 80.8 | 82.2 | 84.4 | **85.6** | 53.4 | 57.2 | 72.8 | **73.5** | 28.3 | 30.7 | 46.5 | **47.7** |
| ● | ○ | ● | ○ | 86.0 | 87.5 | 86.9 | **87.9** | 58.7 | 59.7 | 70.7 | **71.2** | 28.0 | 34.6 | 44.3 | **46.0** |
| ● | ○ | ○ | ● | 83.3 | 85.5 | 87.8 | **88.4** | 67.6 | 72.9 | 82.9 | **84.1** | 68.0 | 70.3 | **77.5** | 77.3 |
| ● | ● | ● | ○ | 85.1 | 86.2 | **88.4** | 88.2 | 70.7 | 74.2 | 83.3 | **84.2** | 69.9 | 71.1 | 75.1 | **76.2** |
| ● | ● | ○ | ● | 87.0 | 88.0 | 87.4 | **88.3** | 61.0 | 61.5 | 67.7 | **67.9** | 33.4 | 34.1 | 42.8 | **43.1** |
| ● | ○ | ● | ● | 87.0 | **88.6** | 87.2 | 88.2 | 72.2 | 75.6 | **82.9** | 82.5 | 69.7 | 71.2 | 73.8 | **75.4** |
| ○ | ● | ● | ● | 82.1 | 83.3 | **86.6** | 86.5 | 70.7 | 75.3 | 83.2 | **84.4** | 69.7 | 71.1 | 75.9 | **76.2** |
| ● | ● | ● | ● | 87.6 | 88.8 | **89.1** | 88.9 | 73.4 | 76.4 | 84.8 | **87.3** | 70.8 | 71.7 | 78.2 | **79.3** |
| Mean | | | | 78.6 | 80.1 | 85.3 | **85.9** | 59.7 | 64.0 | 76.8 | **77.9** | 48.1 | 50.1 | 60.70 | **61.8** |

Figure 3: Visual segmentation results of the SMU-Net on BraTS 2018 using a single modality as an input. The green area shows the WT; red: CT and blue: ET.

Table 2: Contribution of each SMU-Net module on the overall performance

| | $\mathcal{L}_{\text{consistency}}$ | $\mathcal{L}_{\text{context}}$ | $\mathcal{L}_{\text{style}}$ | $style_{\text{module}}$ | WT | CT | ET | Average |
|---|---|---|---|---|---|---|---|---|
| | × | √ | √ | Adversarial | 85.49 | 69.20 | 44.23 | 66.30 |
| | √ | × | √ | Adversarial | 86.41 | 70.38 | 45.14 | 67.31 |
| Flair Modality | √ | √ | × | Adversarial | 86.29 | 69.89 | 45.43 | 67.20 |
| | √ | √ | √ | Distribution | 87.04 | 71.50 | 45.11 | 67.88 |
| | √ | √ | √ | Texture | 86.47 | 70.19 | 45.73 | 67.46 |
| | √ | √ | √ | Adversarial | **87.52** | **71.80** | **46.12** | **68.47** |

## 4. Conclusion

To address the problem of missing MRI modalities in brain tumor segmentation, we offer a unique co-training network that distills the representation of the complete modality model into a missing modality network. Through the use of a style and content matching mechanism our model overcomes common shortcomings of the state-of-the-art, such as loss of modality information. We offer three design choices for the style matching modules and quantitatively analyzed the effectiveness of each module on the missing modality scenario.

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

## Appendix A. Clinical Effect of Missing Modalities

The critical task of brain tumor segmentation could be performed by radiologists but providing annotations in such wise is regarded time-consuming, costly, and, in certain situations, difficult and insufficiently accurate. The author of (Brady, 2017) discusses the causes of current mistakes and discrepancies in radiology. According to (Brady, 2017), both human and system factors may play a role in worsening and deteriorating the radiologist's performance, with a 13% major and a 21% percent minor discrepancy rate for both neuro CT and MR imaging. This intricate and important task may become more difficult if, during multimodal imaging, one or more modalities are missed due to various types of artifacts, since missing these modalities equates to losing critical knowledge that could not be retrieved entirely through other available modalities. The issue of missing modalities is seen as a hurdle not just for radiologists annotated images, but also for automatic segmentation algorithms.

To further emphasize the value and importance of multi modal imaging, particularly for brain tumor segmentation, the case of interest in this study, this section is dedicated to the detailed contribution and relevance of each MRI modality in the BraTS dataset, which was utilized in this work. The BraTS dataset includes a variety of MRI scans (see section 3.1 for further details) with four different modalities: FLAIR, T1, T1c, and T2 with manual annotations of the different sub tissues of the human brain. In this dataset, according to (Menze et al., 2014), four different sub tumor structures are characterized: "edema", "non-enhancing (solid) core", "necrotic (or fluid-filled) core", and "non-enhancing core". Each of the four registered MRI modalities is utilized to differentiate the aforementioned described sub tumor structures in the human brain. In (Menze et al., 2014), this is discussed in great depth and is summarized as follows: the T2 and FLAIR modalities were predominantly used to segment "edema". Hyper-intensity investigation in the T1c modality and visible hypo-intensities in the T1 modality could be used to segment both enhancing and non-enhancing sub regions for high-grade gliomas. The "enhancing core" is segregated by setting an adequate intensity threshold for the T1c modality inside the specified gross tumor core. The visible tortuous and low intensity necrotic structures T1c were interpreted as "necrotic (or fluid-filled) core". And at last, removing the "enhancing core" and "necrotic core" sub regions yields the "non-enhancing (solid) core". Figure 4 shows a sample image from BraTS dataset for all modalities, where each modality reveals distinct characteristics as discussed.

The BraTS dataset divides the four aforementioned sub tissues into three main regions and defines three medically applicable segmentation masks: the "whole" or "complete" tumor that denotes all four sub regions, the tumor "core" that includes "non-enhancing (solid) core", "necrotic (or fluid-filled) core", and "non-enhancing core", and finally, the "active" tumor that encompasses the "enhancing core" regions. Considering the evidence mentioned in (Menze et al., 2014), the conclusion would be that each MRI modality delivers distinct information about different tumor substructures in the human brain in the form of spatial 3D scans. In the absence of each, a radiologist would offer an annotation with less certainty and accuracy, and the performance of most deep learning algorithms designed to do autonomous segmentation would diminish as well. As a result, developing a deep learning network that is robust to missing modalities would be a worthwhile and significant endeavor.

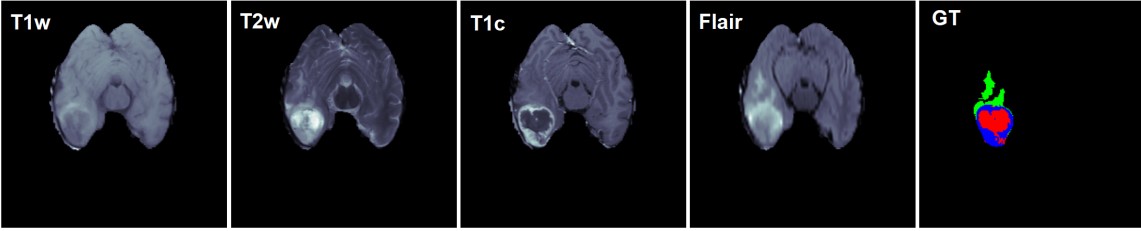

Figure 4: Sample data with different modalities from BraTS dataset. Each modality reveals distinct characteristics of the human brain tissue. In the annotation mask, the Blue area shows GD-enhancing tumour, Green: the peritumoral edema, and Red: tumour core.

## Appendix B. Quantitative Analysis

This section offers a visual resemblance of the data provided in Table 1 in the form of a histogram to include a clearer perspective of SMU-performance. Figure 5 shows the performance of SMU-Net in comparison to two baseline techniques, U-HeMIS (Havaei et al., 2016) and HVED (Dorent et al., 2019), as well as a more recent method, ACN (Wang et al., 2021). The performance comparison is carried out on the BraTS 2018 dataset, using the identical settings for all four evaluated models, with Dice Score as the metric. The quantitative findings reveal that when the number of available modalities during inference time decreases, the performance of HeMIS and HVED approaches largely declines, however this is not the case for our suggested method and the ACN approach, due to their capacity to compensate for missing modalities. In summary, the findings indicate that the knowledge distillation technique, which is utilized in both our method and ACN, improves the network's robustness to missing modalities and is more proficient than the classic common latent space strategy. Our technique, in particular, matches both the content and style representations individually and outperforms ACN in the same settings.

We further discuss the experimental results in the extreme missing modality situation in more detail. More precisely, we offer a benchmark for comparing the performance of different

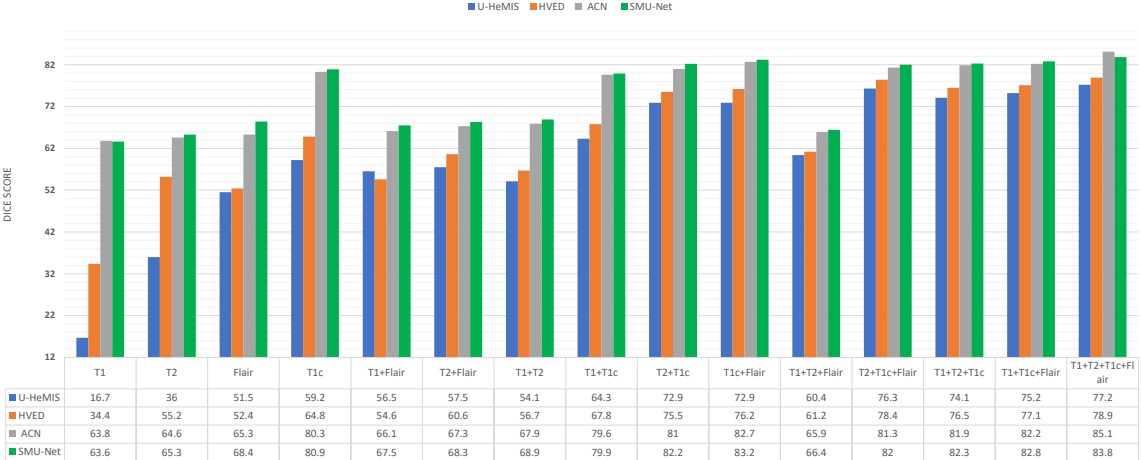

| | T1 | T2 | Flair | T1c | T1+Flair | T2+Flair | T1+T2 | T1+T1c | T2+T1c | T1c+Flair | T1+T2+Flair | T2+T1c+Flair | T1+T2+T1c | T1+T1c+Flair | T1+T2+T1c+Flair |
|---|---|---|---|---|---|---|---|---|---|---|---|---|---|---|---|
| U-HeMIS | 16.7 | 36 | 51.5 | 59.2 | 56.5 | 57.5 | 54.1 | 64.3 | 72.9 | 72.9 | 60.4 | 76.3 | 74.1 | 75.2 | 77.2 |
| HVED | 34.4 | 55.2 | 52.4 | 64.8 | 54.6 | 60.6 | 56.7 | 67.8 | 75.5 | 76.2 | 61.2 | 78.4 | 76.5 | 77.1 | 78.9 |
| ACN | 63.8 | 64.6 | 65.3 | 80.3 | 66.1 | 67.3 | 67.9 | 79.6 | 81 | 82.7 | 65.9 | 81.3 | 81.9 | 82.2 | 85.1 |
| SMU-Net | 63.6 | 65.3 | 68.4 | 80.9 | 67.5 | 68.3 | 68.9 | 79.9 | 82.2 | 83.2 | 66.4 | 82 | 82.3 | 82.8 | 83.8 |

Figure 5: Performance comparison of the proposed SMU-Net methods with the SOTA approaches in more details. Each coloumn of the table shows the performance of the methods only using a mentioned modalities on the inference time.

approaches in the single modality scenario during test phase and full modality scenario throughout the training phase. To that purpose, Table 3 summarizes the quantitative outcomes of several approaches for each modality instance. In addition to the prior networks to which we compared our technique in Table 1 and Figure 5, we include a U-Net baseline method in Table 3, which contains a conventional U-Net model (Ronneberger et al., 2015) with just a single modality as the input data and no mechanism for mitigating the negative impacts of missing modalities to train the network with. Based on the data in Table 3, it is evident that our suggested strategy outperforms the U-Net baseline approach markedly. This finding demonstrates the efficacy of our suggested style-content matching modules in learning rich and generic representation for the task of brain tumor segmentation, in addition to the employed co-training strategy. Furthermore, our technique surpasses ACN in all single modality scenarios and outperforms the U-HeMIS and HVED methods by large margins. Even when compared to the U-Net baseline approach, performance of the U-HeMIS and HVED methods could be considered as insufficient for the extreme missing modality situations, since there methods are not well-designed for a such scenario.

## Appendix C. Details on Network Architecture

Our SMU-Net is comprised of two parallel U-Nets which have the same structure. Our U-Net model follows the regular symmetric structure with four blocks in each encoding and decoding path with two 3D convolutional layers followed by the group normalization and pooling/up-sampling layers. The discriminator of our adversarial style-matching module contains three 3D convolutional blocks each one followed by a ReLU non-linearity and batch normalization layer. It is worth mentioning that we trained our model in an end-to-end manner using Adam solver with learning rate of $10^{-4}$ for 250 epochs with batch-size

Table 3: Experimental results of the literature work on BraTS series with extreme missing modality scenario. We use the average of whole, enhanced and core tumor segmentation scores to report the dice score for each modality.

| Article | Dice score | | | | |
|---|---|---|---|---|---|
| | **T1** | **T1c** | **T2** | **FLAIR** | **AVG** |
| U-Net Baseline | 51.2 | 68.2 | 52.3 | 57.6 | 57.3 |
| U-HeMIS | 16.7 | 59.2 | 36.0 | 51.5 | 48.8 |
| HVED | 34.4 | 64.8 | 55.2 | 52.4 | 51.7 |
| ACN | 63.8 | 80.3 | 64.6 | 65.3 | 68.5 |
| SMU-Net | 63.3 | 80.9 | 65.3 | 68.4 | **69.4** |

of 1. We should note that the segmentation mask is obtained from both streams of the network in the training phase. In contrast, we merely utilize the mask acquired from the missing modality path in inference time.

## Appendix D. Neural Style Transfer Motivation

The challenge of transferring the texture of one picture to another while keeping the semantic information of the second image is known as style transfer (Gatys et al., 2016). Reconstructing or synthesizing the texture of an artistic painting or a natural and photo-realistic image has been an interesting task in the field of computer vision and image processing, with applications including video and image compression, image denoising, and occlusion fill-in (Efros and Leung, 1999; Wei and Levoy, 2000). The idea of style synthesis and transfer was further developed in (Gatys et al., 2015). They presented the first deep neural networks algorithm that can dissect an image's characteristics into a content and style representation in 2015. They were able to demonstrate through experiments and remarkable results, using a novel neural algorithm, that the content and the style of an image could be extracted independently and then also recombined. They pointed out that convolutional neural networks, which are trained to detect objects in pictures will primarily observe and extract the actual content and the overall layout of the existent items. To put it another way, reconstructing a picture from the feature maps retrieved in each layer results in a representation that mostly projects the content of the input image. The picture's information, such as the objects and their locations in the image, as well as their arrangements, is acquired from the network's higher layers as content information. A representation that simply takes into account the high-level information would have a considerable difficulty reconstructing the detailed value for each pixel. The authors of (Gatys et al., 2015) built a feature space for retaining the style information, in order to maintain the precise pixel values and produce a representation that conveys information that is mostly about color, saturation, and texture. The style could be extracted from a few low-level layers in the networks, resulting in a more local representation. On the other hand, the authors of (Gatys et al., 2015) advocate leveraging numerous layers of the network to build a global multi-scale style representation.

The effectiveness of neural style transfer was also utilized in deep learning networks that process MR images. The authors of (Denck et al., 2021), for example, assert that

contrast differentiation across MRI modalities would result in a style variation, therefor, they presented an image-to-image generative adversarial network capable of synthesizing MR pictures with configurable contrast via style transfer. The network presented by the authors of (Tomar et al., 2022) also employ a style encoder to group images with roughly similar styles together. According to (Ma et al., 2019), utilizing neural style transfer to overcome existing inconsistencies in brightness, contrast, and texture, all of which are considered style inconsistencies, the network's segmentation performance would enhance. The SMU-Net style transfer technique is based on the work of the authors of (Gatys et al., 2015), which was previously discussed in this section. We suggest decomposing feature representations into a style and a content representation, which, to the best of our knowledge, has not been explored on MRI scans, particularly in the situation of brain tumor segmentation with missing modalities.

## Appendix E. A Detailed Clarification on Style Matching Module

As stated in the paper, to perform the matching mechanism, we defined three different style matching modules namely, distribution matching, texture matching and adversarial matching. Moreover, we suppose that the modification module ( figure 1) adjusts the style of the missing modality feature map based on one of the three aforementioned modules. In this section, we will provide further intuition on the concept behind each one.

**Distribution Matching Module**: Once trained, on the inference a resampling method applies to obtain a sample from the latent space to adjust the style representation. To this end, the acquired sample will be concatenated with the style encoder feature map to reach the aforementioned goal of approximating the missing modality style in subject to the full modality one.

**Texture Matching Module**: In this case, guided by the texture matching loss, we learn to change the missing modality style.

**Adversarial Matching Module**: Here, the adversarial loss itself will gain an understanding of how to change the style of the missing modality according to the full modality style. A good example of this intuition is the (Zhang et al., 2018) paper.

All aforementioned modules are shown in figure 1 with "Style Matching Module", which trains based on the related loss function. Eventually, the matching module produces a modification function to modify the style of the missing modality path. This function is highlighted in the figure which only applies to the missing modality path. Besides these modules, as we depicted in the figure, we devised the style loss (same as texture loss) to further ensure the style matching procedure. For the sake of more clarification, we should note that in the case of using a texture matching module we are needless to modification the function of the style matching module as the texture loss itself guarantees the style matching goal. In contrast, while exploiting the two other matching modules, specifically adversarial and distribution, the style loss will perform the style matching mechanism.

