# OpenReview forum: "SMU-Net: Style matching U-Net for brain tumor segmentation with missing modalities"
_MIDL.io/2022/Conference — MIDL 2022_

### Official Review · Reviewer_s7QJ · 2022-01-20

**Confidence:** 4
**Preliminary Rating:** 3
**Recommendation:** Poster

**Summary:**

The authors propose a pipeline to mitigate a missing MRI modality issue for brain tumor segmentation. For that, they use a network composed of two separate subparts that input different information. The first subpart inputs all four MRI modalities (available from the BraTS dataset), while the second one takes only a subset of the modalities (less than four). Throughout the depth of both subparts, there are various feature matching components that aim to distill the representation learned by the first subpart into the representations of the second subpart.
This is a potentially clinically useful area of research as not always all four modalities are given.

**Strengths:**

For mitigating the missing modality issue, the authors essentially take different deep-learning alchemy ingredients (e.g. adversarial learning, style-matching, distribution matching) and boil a magic soup thereof. The authors claim that the soup has the benefit of bringing closer feature representations between a network subpart that takes all four MRI modalities as input and another subpart that inputs less than four modalities.
The method is compared with existing approaches for the missing modality problem and shows a convincing performance increase.

**Weaknesses:**

There are typos and logical fallacies in every mathematical definition (see below), and the method is a combination of multiple existing elements of deep-learning machinery. Especially, after reading the comments from another reviewer, who pointed to the existent highly-similar work by Wang et al. 2021, I fully share concerns of the reviewer that the work by Wang et al. is exactly the paper that the proposed method should be compared to. Given also that the contributions of the proposed method compared to the method in Wang et al. are minor, I lean towards "Borderline".

**Deanonymize Review:**

no

**Detailed Comments:**

There are some issues with the math here:
- in Eq.1. First, you do not define what are lowercase x_f and x_m (even though I can guess what are those, it is important to mention them in the text). Next, your definition of the Jenson-Shanon estimator is clearly different from the one provided in Belghazi et al., 2018. In the latter, the first term corresponds to the supremum of expectations on the learnable function, and the second term corresponds to the log of expectation on the exponentiated learnable function. Thus you should either correct or explain why your definition is different.
- in Eq.2 things can be more accurate as well. You say that you compute L1 between soft logits, but you introduce new notations for the soft logits, different notations from Eq.1, where you also use soft-logits. Please make notations for soft-logits consistent between Eq.1 and Eq.2. Also, I do not understand where GP, the global polling, coming into the game. You compare here soft-logits, which are the output of the co-training network, CT, right? If so, then for the sake of clarity in Eq. 2 you should use CT instead of GP.
- In Eq.3, it does not seem that under the function T the argument f_s_m is needed since f_s_m is already included in f_s_m. Also, in all your figures you denote f_s_m as f_s_s. Please be consistent, choose one notation. Please say that z - is a sample from the continuous "Gaussian" distribution. Please say what is T and what is P (again even though I can guess, it requires mentioning).
- In Eq.4, it is misleading to use superscripts h and w, as they can be confused with the height and width of the features. But I doubt that you index the height and width. I assume you index here a whole feature and show that you sum over the channel dimension, so please use something like "c" here for indexing.
- In Eq.5, please say what is w_l. I think the argument of the L_style should be f_s_f, and f_s_m (or f_s_s for the latter).
- In Eq.6, seems to be correct :) Just please decide whether it is f_c_m or f_c_s.

**Final Rating After The Rebuttal:**

4: Weak Accept

**Justification Of The Final Rating:**

Even though the improvement is marginal ~1%, most of my comments are addressed, so I increase my score to "weak accept".
Even though the improvement is marginal ~1%, most of my comments are addressed, so I increase my score to "weak accept".

**Paper Type:**

both

**Questions To Address In The Rebuttal:**

Please fix the mathematical issues as pointed out in the "Detailed comments" section, and comment on the concerns raised by another reviewer on comparison with Wang et al. Please also comment on whether it is crucial to keep the weights of the decoder networks shared. Also, what predicted mask do you take for computing DICE? From the subnetwork taking all four modalities as input or the other subnetwork?

**Special Issue:**

no

---

### Official Review · Reviewer_Bmrx · 2022-01-22

**Confidence:** 4
**Preliminary Rating:** 2
**Recommendation:** Poster

**Summary:**

In their paper, the authors tackle the problem of brain tumor segmentation with missing resp. subsets of available MRI modalities compared to a full set of modalities (T1, T1c, T2, FLAIR), using a convolutional neural network model (U-net). As key components of their method, they introduce a co-training strategy between the two scenarios (full set / reduced set / single modality). This is supported / enabled by the decomposition of either input into a "content" and "style" representation, which are then enforced to be similar by a cost function guiding the training of the neural network. A decoder with shared weights then generates a prediction mask, for which a further consistency loss is applied. The approach is evaluated on the segmentation dataset from the 2018 BraTS challenge (Glioma, complete, core & enhancing tumor) and compared with results of two approaches (U-HeMIS, HVED) on the same datasets. The algorithm performs favourably compared to the included state-of-the-art with improvements especially for the single-modality case.

**Strengths:**

- The idea to combine co-training and use a "style"-/"content"-approach to align shallow and deep feature representations somewhat independently is interesting and the authors propose three different loss functions for the style-approach.
- The authors evaluate their approach on a publicly available data set following reproducible data splits and compare the proposed method to two previous approaches. An initial ablation study investigates the contribution of different parts of their proposed method.
- The authors plan to make their source code publicly available after acceptance.

**Weaknesses:**

- The difference to adversarial co-training described in Wang et al. 2021 is not described, but instead the similarities (both are co-training approaches with multi-component representation alignment strategies) seem to be brushed over in the related work section. The results of Wang et al. are also not included in the state-of-the-art. This should be clarified and (additional) ablation studies should be done to investigate/validate the benefits of the different modules proposed in either approach
- The notation in the paper is not very consistent and rather confusing, both in terms of the used symbols and the notation used in the figure vs. the notation used in the description. Examples: ' stands both for missing modalities (M') and once for prediction (Y') independent of modalities; use of "s" for both style and single; unclear mixing of "missing" and "single" modality case, unclear definition of the ground truth (binary representation?)
This should be re-worked and a consistent notation should be introduced.
- Several details of the method are not clearly described, e.g., network architecture of backbone, decoder, adversarial network, resp. training strategies, what is the yellow box in Fig 1? The authors first talk about "missing modalities" and then about "single modality networks" - Is a pair of networks trained for each combination of possible missing modalities? Are only uni-modal networks trained and the predictions combined? If so, how? Would/wouldn't it make sense to train *all* networks jointly?
- The description and justification of the style and content separation is not clear to me. On the one hand, the authors state: "[...] the style matching function aims to reconstruct the missing information using the full-modality distribution while the content matching module tries to preserve the modality-specific information." In the abstract, it rather sounds as if the idea is to align both early and deep representations ("Our style matching module [...] to transfer the feature from full-modality path [...] our content module re-calibrates [...] based on discriminative semantic features. The derived style representation (early + deep) maintains valuable textural information, whereas the content feature (only deep) contains the image's core structural and semantic characteristics.")


**Deanonymize Review:**

yes

**Detailed Comments:**

- Additional baseline and ablation studies would strengthen the paper considerably: What is the performance if no alignment is used and the networks are simply trained on the resp. missing modality setting? Why is only a combination of modules evaluated in Table 2?
- Given the number of experiments performed, analysing the variability of the results with repeated trainings would be very interesting to understand better whether a difference of e.g., 0.5 percentage points constitutes a true improvement or is rather a random effect
- The performance of the missing modality cases seems to strongly depend on the best performing available modality (not very surprising). This should be discussed.
- While the space is certainly limited, additional details would be a good addition in the supplementary material (e.g., network architectures)
- It is unclear which style module was used for the results in Table 1. What would happen if multiple modules used simultaneously?
- The description of the L1-loss was not clear to me, with the global pooling, is a single value aligned here? How exactly is the consistency loss composed? Here, also an ablation study might be interesting.

Clarity of the description of related work could be improved, e.g.:
- p.2: "[...] than the classifier" - which classifiers do the authors refer to? The description of the related work by Van Tulder + de Bruijne is not clear. What is the role of the classifier in the segmentation?
- While the authors mention and reference HeMIS / HVED, one aspect of the motivation of these approaches does not become clear, i.e., the possibility to use a flexible / shared approach with varying number and combination of inputs (see also weaknesses).
- Also the statement "[...] usually performs poorly when more than one modality is missing." seems to be very general given that the performance (for all approaches) very strongly depends on *which* modalities are missing
- criticism of works by Zhou et al. is not clearly explained. What do the authors mean by "not regarded a fully fulfilling method"?
- why have one out of three, which not all?

Minor comments:
- Short title (header of the pages) is currently not available
- Spelling of the BraTS challenge; full challenge should be mentioned & referenced somewhere
- Introduction seems to be a bit hand-waving and can potentially be clarified. Are there references for the initial part of the introduction? What is the motivation for having the tumor subregions? What does it help / aid? Also:
. "molecular structure" - maybe a bit too fine-grained
. "information [...] for diagnosis and cancer prevention." - why / how prevention?
. "Analyzing these medical data requires ..." - a bit unclear
- p.2: "[...] to use deep learning power" - unclear
- p.6: "along with the reconstructed ones," - unclear
- p.6: "learning rate 10 - 4" - typo
- p.7: "SMU-Net achieves considerably superior results." - imprecise

**Final Rating After The Rebuttal:**

4: Weak Accept

**Justification Of The Final Rating:**

The authors extensively commented on the concerns raised by the reviewers, and while the improvements are rather marginal compared to the SOTA, the authors rather extensively evaluate different modules.

**Paper Type:**

both

**Questions To Address In The Rebuttal:**

- What do the authors see as the main contribution compared to Wang et al. 2021, and how would you validate the benefit of your extension?
- What is the motivation of calling the two modes "style" and "content" (apart from the fact that this is the name in related work)? Can it be made more clear what the connection to MRI images is? (How) is it prevented that style also contains content information if also deep features are included?
- How is the missing modality case handled? E.g., if two modalities are available, are the single-modality predictions combined? Is there a network (co-)trained for each possible combination of modalities?

**Special Issue:**

no

---

### Meta-Review · Area_Chair_Uqmn · 2022-02-14

**Recommendation:** Accept (Poster)
**Confidence:** 4

**Metareview:**

This paper proposes a meta-architecture to deal with missing modalities in medical image segmentation. While the improvements with respect to the SOTA are marginal, the experiments are extensive and the method is certainly of interest to the MIDL audience -- as all reviewers (including me) agree.

---

### Decision · Program_Chairs · 2022-02-28

Accept